# Hypoxia-Reoxygenation Impairs Autophagy-Lysosomal Machinery in Primary Human Trophoblasts Mimicking Placental Pathology of Early-Onset Preeclampsia

**DOI:** 10.3390/ijms23105644

**Published:** 2022-05-18

**Authors:** Shibin Cheng, Zheping Huang, Sukanta Jash, Kathleen Wu, Shigeru Saito, Akitoshi Nakashima, Surendra Sharma

**Affiliations:** 1Department of Pediatrics, Women & Infants Hospital of Rhode Island, Warren Alpert Medical School of Brown University, Providence, RI 02905, USA; zhuang@wihri.org (Z.H.); sjash@wihri.org (S.J.); kathleen_wu@brown.edu (K.W.); ssharma@wihri.org (S.S.); 2Department of Obstetrics and Gynecology, Faculty of Medicine, University of Toyama, Toyama 30-0194, Japan; s30saito@med.u-toyama.ac.jp (S.S.); akinaka@med.u-toyama.ac.jp (A.N.)

**Keywords:** autophagy-related proteins, autolysosome, autophagosome, electron microscopy, preeclampsia, proteasome, trophoblasts

## Abstract

We have previously described that placental activation of autophagy is a central feature of normal pregnancy, whereas autophagy is impaired in preeclampsia (PE). Here, we show that hypoxia–reoxygenation (H/R) treatment dysregulates key molecules that maintain autophagy–lysosomal flux in primary human trophoblasts (PHTs). Ultrastructural analysis using transmission electron microscopy reveals a significant reduction in autophagosomes and autolysosomes in H/R-exposed PHTs. H/R-induced accumulation of protein aggregates follows a similar pattern that occurs in PHTs treated with a lysosomal disruptor, chloroquine. Importantly, the placenta from early-onset PE deliveries exhibits the same features as seen in H/R-treated PHTs. Taken together, our results indicate that H/R disrupts autophagic machinery in PHTs and that impaired autophagy in the placenta from early-onset PE deliveries mimics the events in H/R-treated PHTs. Notably, assessment of key regulators at each stage of autophagic processes, especially lysosomal integrity, and verification of autophagic ultrastructure are essential for an accurate evaluation of autophagy activity in human trophoblasts and placental tissue from PE deliveries.

## 1. Introduction

Macroautophagy/autophagy (autophagy) is the source of the catabolic activity of the cell for clearing dysfunctional cytoplasmic components and damaged organelles [1,2,3,4,5,6]. Autophagy plays a critical role in cell survival, differentiation, aging, and homeostasis [1,2,3,4,5,6]. Prior studies have described a crucial regulatory role for autophagy in normal pregnancy maintenance and protection against pathogens and stress [7,8,9,10,11,12,13,14,15,16,17]. We have shown that impaired autophagy is associated with the pathogenesis of preeclampsia (PE), a multi-organ, pregnancy-specific syndrome, and a leading cause of maternal and prenatal morbidity and mortality [11,12,13,14,15]. PE is diagnosed by de novo onset of hypertension at or after 20 weeks of gestation with subtypes of early- (e-PE, < 34 weeks of gestation) and late-onset PE (≥ 34 weeks of gestation) [18,19,20,21,22,23,24]. PE affects 3–8% of all pregnancies and poses a risk of many chronic diseases, including cardiovascular disease, diabetes mellitus, kidney injury, and neurodegenerative diseases in mother and offspring later in life [25,26,27,28,29]. To date, there is no effective therapy available for this severe pregnancy complication. Delivery of the placenta is the most effective intervention but may contribute to the incidence of preterm birth.

Despite extensive studies, the pathogenesis of PE remains poorly understood. It is well accepted that the PE etiology entails a wide range of pathophysiological processes, including hypoxia/ischemia, endoplasmic reticulum (ER) stress, oxidative stress, pyroptosis, inflammation, and release of anti-angiogenetic factors [15,18,19,20,21,22,23,24,25,26,27,28,29,30,31,32,33,34,35,36,37,38,39,40]. We and others have recently demonstrated that the etiology of PE is associated with protein misfolding and aggregation, a hallmark of neurodegeneration diseases such as Alzheimer’s disease [12,13,41,42,43]. We have demonstrated that lysosomal biogenesis proteome defect and autophagic impairment contribute to the accumulation of aggregated proteins in the placenta from PE deliveries as well as in a mouse model of PE [12,13].

The autophagy machinery entails a cascade of processes that involve isolation membrane initiation, cargo recognition/recruitment, phagophore formation, autophagosome generation, fusion of the autophagosome with the lysosome (autolysosome formation), and degradation of cargoes in the autolysosome (Figure 1A). Each step of autophagic flux is tightly regulated by a myriad of evolutionally conserved proteins encoded by the autophagy-related genes (Figure 1A). However, which regulatory molecules in the autophagy–lysosomal pathway are dysregulated in the placenta or the cellular models of PE remain poorly understood [44,45,46,47,48,49,50,51,52,53].

The aim of this study is to determine regulatory molecules in autophagic flux that are dysregulated in the PE placenta and a cellular model of PE. Since chronic low oxygen tension has been associated with the pathophysiology of PE [12,18,38,39,40,54], a cellular model that comprises exposure of primary human trophoblasts (PHTs) to ER inducer, hypoxia–reoxygenation (H/R) is used to mimic PE pathophysiology. Here, we systematically investigated the autophagy–lysosomal machinery by examining key molecules at each step of the pathway in either H/R-treated PHTs or the placenta from women withe-PE. Our results suggest that exposure of PHTs to H/R disrupts the autophagy–lysosomal machinery by selectively affecting the physiological abundance of several key regulatory molecules and protein complexes. These results were validated by compelling ultrastructural profiles of normoxia and H/R-treated PHTs. Importantly, the e-PE placenta displays a similarly reduced expression of some of the same key regulatory molecules that promote autophagy activity.

## 2. Results

### 2.1. H/R Exposure Dysregulates the unc-51-like Autophagy Activating Kinase (ULK1) Regulatory Complex and the Class III Phosphatidylinositol 3-Kinase (PI3K-III) Complex in PHTs

As depicted in the schematic diagram in Figure 1A, isolation membrane nucleation is initiated by the ULK1 complex, including ULK1, Atg13, and Atg101, whereas membrane trafficking is regulated by the PI3K-III complex, including Beclin 1 and Atg14L [6,55,56,57]. To evaluate these molecules in PHTs, we assessed the expression levels of ULK1, Atg13, Atg101, Beclin 1, and Atg14L proteins using Western blotting. PHTs were subjected to H/R or normoxia for 3 days as described in Materials and Methods. The relative abundance of all target proteins was calculated by normalizing the intensity of each protein band to β-actin band. Immunoblotting and quantitative analyses showed that H/R stimulation significantly downregulated the abundance of ULK1, Atg13, Atg101, and Atg14L, but did not alter Beclin 1 expression as compared to normoxia (Figure 1B,C).

### 2.2. H/R Treatment Upregulates Phosphorylated mTORC1 and UV Radiation Resistance Associated Gene Product (UVRAG) in PHTs 

mTORC1 is a master negative regulator of autophagy [6,54,57]. Prior studies have demonstrated that the ULK1 initiation complex can be inhibited by mTORC1, and that mTORC1 can negatively regulate autophagosome and endosome maturation through a complex containing UVRAG [6,55,58]. The abundance of mTORC1 and UVRAG was then examined in H/R-treated PHTs using immunoblotting. As shown in Figure 1D–F, exposure to H/R significantly augmented the contents of phosphorylated mTORC1 and UVRAG relative to normoxia-treated PHTs.

### 2.3. H/R Exposure Significantly Alters the Content of Several Key Autophagy Proteins, including Atg12-Atg5 Conjugation Complex, LC3, and GABARAP

Studies have shown that autophagosome formation is regulated by a ubiquitin-like conjugation system in which Atg12 is covalently bound to Atg5 (referred to as Atg12-Atg5 conjugate) and targeted to autophagosome vacuoles [59]. This conjugation reaction can be mediated by the ubiquitin E1-like enzyme Atg7 [59,60,61,62]. The Atg12-Atg5 conjugate plays an essential role in the formation of isolation membrane, phagophore, and autophagosome [59,60,61,62]. Therefore, we next examined the presence of Atg5, Atg12, and Atg7 in PHTs exposed to H/R. Our results showed a significant decrease in Atg12-Atg5 expression, but no significant alteration was observed in Atg7 in H/R-treated cells compared to normoxia-exposed cells (Figure 2).

Pro-forms of LC3 and its homologue GABARAP are hydrolyzed by a cysteine protease, Atg4B, which generates LC3-I and GABARAP-I, respectively [63,64,65]. LC3-I and GABARAP-I become LC3-II and GABARAP-II, respectively, through conjugation to the head group of the lipid phosphatidylethanolamine catalyzed by a series of ubiquitin-like reactions, which involves a number of enzymes, such as Atg3, Atg7, and the Atg12-Atg5 conjugate [63,64,65] (Figure 1). LC3-II and GABARAP-II play critical roles in cargo recruitment [63,64,65]. To determine the effect of H/R exposure on cargo recognition and recruitment for autophagosome formation, we then evaluated the expression of Atg3, LC3, and GABARAP. As shown in Figure 2, Atg12–Atg5, Atg3, LC3-II, and GABARAP-II were significantly attenuated in PHTs exposed to H/R. Rab11, a small GTPase that specifies recycling endosomes, plays a critical role in autophagosome assembly via trafficking autophagy proteins such as Atg14 and Beclin 1 [66]. Next, we investigated the abundance of Rab11 and found that H/R treatment did not significantly alter the content of this protein when compared to normoxic condition (Figure 2).

### 2.4. Exposure to H/R Decreases the Expression of LAMP1, LAMP2, Cathepsin B and D, and LAMTOR4, and Increases the Abundance of Rubicon and Inhibits Lysosomal Activity in PHTs

Degradation of cargoes loaded in autophagosome depends on the fusion of the autophagosome with the lysosome which harbors various proteases that can digest autophagic cargoes [1,2,3,4,5,6]. Prior studies have demonstrated that a regulatory protein complex involving Rubicon can inhibit the fusion of the autophagosome with the lysosome and trafficking of the autolysosome [67,68,69,70,71,72,73,74]. LAMTOR4 is a component of the regulatory complex that is essential for the functional integrity of the lysosome [75]. Here, we evaluated the influence of H/R exposure on the lysosomal integrity and regulators for autolysosome formation through examining the expression of LAMP1, LAMP2, cathepsin B and D, LAMTOR4, and Rubicon. Our results indicated that exposure to H/R significantly reduced the abundance of LAMP1, LAMP2, cathepsin B and D, and LAMTOR4, but elevated Rubicon expression as compared to normoxic treatment (Figure 3A–G). The LysoTracker dye, a highly soluble small molecule that freely permeates cell membranes and selectively stains acidic subcellular organelles such as lysosomes and autolysosomes, has been widely utilized to detect autophagy-associated lysosomal activity [12]. To reinforce the above findings, we examined the lysosomal activity using LysoTracker dye. As demonstrated in Figure 3H,I, H/R-treated PHTs exhibited significantly lower levels of LysoTracker signal as compared to normoxia-treated cells, suggesting that the lysosomal activity is inhibited in PHTs exposed to H/R.

### 2.5. Chloroquine Treatment and H/R Exposure Induce the Accumulation of Protein Aggregates in PHTs

The autophagy–lysosome pathway plays a crucial role in the clearance of protein aggregates and damaged organelles [1,2,3,4,5,6]. Impaired autophagy–lysosome machinery has been reported to contribute to the accumulation of protein aggregates in a number of diseases such as Alzheimer’s disease and PE [12,13,41]. Therefore, if H/R treatment disrupts autophagic flux, degradation of protein aggregates should be compromised, and as a result, aggregated proteins should be accumulated in H/R-exposed trophoblasts. To test this, we determined whether H/R exposure induces protein aggregation in PHTs using ProteoStat, a dye that uniquely binds aggregated proteins. We have recently developed a novel method of detecting ProteoStat-positive protein aggregates in serum from Alzheimer’s disease (AD) and PE patients [76]. Chloroquine is known to impair lysosomal acidification and consequently disrupt autolysosome formation [18,77]. Thus, chloroquine treatment was used as a positive control in parallel experiments. Chloroquine remarkably induced the accumulation of aggregated proteins as evidenced by robust ProteoStat fluorescence (Figure 4). Intriguingly, in a similar fashion as chloroquine treatment, H/R but not normoxia exposure elicited a robust accumulation of protein aggregates (Figure 4). This result supports the notion that exposure to H/R disrupts autophagy–lysosomal machinery in PHTs.

### 2.6. Ultrastructural Analysis of Autophagy Morphology in PHTs Exposed to H/R Condition

To validate the observations described above and to characterize the ultrastructural morphology of autophagic vacuoles in H/R-treated trophoblasts, we employed transmission electron microscopy (TEM). Under TEM, autophagosome and autolysosome have distinct ultrastructural features. For example, the autophagosome appears as the double-membrane structure containing the cytoplasmic contents and/or damaged organelles, while the autolysosome is characterized by a single-membrane vacuole containing electron-dense lysosomal materials and cytoplasmic contents and/organelles at various states of degradation. Ultrastructural images and semi-quantitative analysis of H/R-exposed PHTs unveiled a significantly lower number of autophagosomes and autolysosomes compared to normoxia treatment (Figure 5A–D, *p* < 0.01). Furthermore, H/R-treated cells exhibited a large number of phagophores and isolation membranes (Figure 5B). These findings demonstrated that exposure to H/R inhibited the formation of autophagosomes and autolysosomes, which was consistent with observations obtained from immunoblotting experiments.

### 2.7. Expression of Atg101, Atg3, Atg16L1, Atg12–Atg5, and GABARAP Was Reduced in the Placenta from e-PE Deliveries

To determine whether molecular events we observed in H/R-stimulated trophoblasts can be recapitulated in the PE placenta, we assessed the key regulators of autophagosome formation in the placenta from women with e-PE and respective gestational age-matched pregnancies. As shown in Figure 6, e-PE placentas contained significantly reduced contents of Atg12–Atg5, Atg16L1, Atg3, Atg101, and GABARAP proteins but comparable levels of UVRAG and LAMTOR4 proteins relative to control samples (Figure 6A,B,H, *p* > 0.05).

## 3. Discussion

Autophagy plays an essential role in embryogenesis, placentation, and normal pregnancy maintenance [15,16,17]. Autophagic activation is a key feature of normal pregnancy and protection against viral infections in the placenta [78]. We have previously shown that autophagy is impaired in PE. However, how autophagy is dysregulated in PE remains controversial. Placental hypoxia/ischemia has been considered a crucial trigger contributing to the onset of PE [37,38,39,40]. Thus, in this study, we employed a cellular model mimicking hypoxia-associated PE pathology. The cellular model using H/R-exposed PHTs was employed to investigate key molecules mediating each step of the autophagy machinery. We compared the observations from this model with the data obtained from the placenta from e-PE patients. We show that the autophagy was impaired in both H/R-exposed PHTs and e-PE placenta. Below, we provide mechanistic explanations for these observations and other intriguing findings. 

Autophagic activation starts with the initiation of isolation membrane formation, which is regulated by the ULK1 complex containing ULK1, Atg101, and Atg13; this is followed by cargo recruitment and phagophore and autophagosome formation, which are promoted by the Atg12-Atg5 and LC3/GABARAP conjugation complexes [56,57,58,59,60,61,62,63,64,65]. Our results demonstrated a significant decrease in the abundance of these regulators including ULK1, Atg101, Atg13, Atg12–Atg5, and LC3/GABARAP, suggesting the disruption of isolation membrane formation, cargo recruitment, and phagophore and autophagosome formation in H/R-treated cells. 

At the final stage of autophagic flux, the autophagosome fuses with the lysosome to form the autolysosome, the organelle where cargoes are degraded by lysosomal proteases [1,2,3,4,5,6]. As such, the lysosome plays an essential role in autophagic degradation machinery. In this regard, our results showed that H/R stimulation not only downregulated the lysosomal resident proteins, including LAMP1, LAMP2, cathepsin D, and cathepsin B, but also reduced the abundance of the lysosomal regulator, LAMTOR4. LAMTOR4 is a component of the Rag-Ragulator complex, a key regulator promoting the lysosome in microglia [75]. Consistently, we found that H/R exposure inhibited the lysosomal activity using LysoTracker staining. These data suggest that H/R disrupts lysosomal integrity and activity, which is consistent with our prior studies showing that exposure to low oxygen tension (<1% O_2_) condition dysregulates TFEB expression in PHTs [12]. TFEB is the master regulator of lysosomal biogenesis machinery. 

Prior studies have shown that some proteins or their complexes can affect autophagic flux at both early and late stages of autophagy activation, such as Beclin 1-composed nucleation complex and mTORC1. Beclin 1 is a mammalian ortholog of the yeast autophagy-related gene 6 (Atg6). Beclin 1 coupled with class III phosphatidylinositol-3-kinase (PI3K) forms distinct complexes with Atg14L, UVRAG, and Rubicon. The Atg14L-containing Beclin 1-PI3K complex positively regulates autophagy at early stages by promoting the formation of isolation membrane, phagophore, and autophagosome [56,57,69,70,71,72,73,74]. In contrast, the Rubicon-UVRAG-containing Beclin 1-PI3K complex impairs the late endosomal/lysosomal structures and consequently inhibits autolysosomal formation [56,57,69,70,71,72,73,74]. mTORC1 not only inhibits phagophore formation at the early stage of autophagic flux by phosphorylating ULK1, but also suppresses autolysosomal maturation at later stages [55,58]. Our results demonstrate that H/R exposure leads to reduction in Atg14L and upregulation of mTORC1, UVRAG, and Rubicon, albeit no effect on the abundance of Beclin 1. Since Rubicon, a negative regulator of autophagy, is associated with an age-dependent decline in autophagy activity and the process of aging [69], we propose that placental aging may be a key feature of the PE placenta. Taken together, these data indicate that H/R exposure disrupts autophagic flux not only through negatively affecting isolation membrane initiation, cargo recognition/recruitment, phagophore and autophagosome formation, and late endosome/lysosome trafficking, but also by dysregulating lysosomal integrity and autolysosome formation. In support of this, we found that H/R induces the accumulation of robust protein aggregates in a pattern similar to that when autophagy is disrupted by chloroquine in PHTs. Most importantly, these results were validated by our ultrastructural data showing decreased numbers of autophagosome and autolysosome in H/R-exposed PHTs using TEM.

Notably, H/R treatment does not influence all the molecules we examined. For example, no significant alteration in the contents of Beclin 1, Atg7 and Rab11 proteins was observed in PHTs in response to H/R stimulation. Based on these results, it is suggested that studies merely on the above molecules may lead to controversial conclusions regarding autophagy activation or impairment in PE. These observations may explain why contradictory results have been reported regarding how hypoxia affects autophagy except for reasons that are attributed to difference in the kinetics of hypoxic treatment and cell types [51,53,79]. 

Based on the results from our cellular model of PE, we interrogated whether multiple key molecules that were significantly influenced by H/R are also subjected to alteration in the placentas from e-PE. We found a remarkable impairment of the autophagy–lysosomal pathway in the placenta from e-PE compared to control placental tissue. Interestingly, the increase in the abundance of UVRAG and LAMTOR4 in H/R-exposed PHTs was not observed in the PE placental tissues. This suggests that the pathophysiological paradigms in the placenta from e-PE patients involve other molecular mechanisms in addition to chronic hypoxia. 

In conclusion, our results provide compelling evidence for the impairment of autophagy–lysosomal machinery in human trophoblasts in response to H/R and in the placenta from women with e-PE (Figure 7). It is necessary to extensively examine regulators at each step of autophagic flux, especially lysosomal activity, and validate the results by analyzing the ultrastructural details of autophagic vacuoles.

## 4. Materials and Methods

### 4.1. Antibodies and Reagents

The following antibodies (Ab) were used: anti-β actin (1:10,000; Abcam, ab6276, Waltham, MA, USA), anti-LAMP1 (1:1000; Abcam, ab25630, Waltham, MA, USA), anti-LAMP2 (1:1000; Abcam, ab25631, Waltham, MA, USA), Atg12 (D88H11) Rabbit mAb (1:1000; Cell Signaling Technology, #4180, Danvers, MA, USA), Atg5 (D5F5U) Rabbit mAb (1:1000; Cell Signaling Technology, #12994, Danvers, MA, USA), Atg3 Rabbit mAb, (1:1000; Cell Signaling Technology, #3415, Danvers, MA, USA), Atg7 (D12B11) Rabbit mAb, (1:1000; Cell Signaling Technology, #8558, Danvers, MA, USA), Atg14 Antibody (1:1000; Cell Signaling Technology, #5504, Danvers, MA, USA), ULK1 (D8H5) Rabbit mAb (1:1000; Cell Signaling Technology, #8054, Danvers, MA, USA), Atg101 (E1Z4W) Rabbit mAb (1:1000; Cell Signaling Technology, #13492, Danvers, MA, USA), Atg13 (D4P1K) Rabbit mAb (1:1000; Cell Signaling Technology, #13273, Danvers, MA, USA), Cathepsin B (D1C7Y) XP^®^ Rabbit mAb (1:1000; Cell Signaling Technology, #31718, Danvers, MA, USA), Cathepsin D Antibody (1:1000; Cell Signaling Technology, #2284, Danvers, MA, USA), Rubicon (D8B2) Rabbit mAb (1:1000; Cell Signaling Technology, #7151, Danvers, MA, USA), GABARAP (E1J4E) Rabbit mAb (1:1000; Cell Signaling Technology, #13733, Danvers, MA, USA), LAMTOR4/C7orf59 (D4P6O) Rabbit mAb (1:1000; Cell Signaling Technology, #13140), anti-MAP1LC3B (1:1000; MBL, PM036). The following horseradish peroxidases (HRP)-conjugated secondary Abs were used: anti-mouse/rabbit IgG-HRP conjugate (1:1000; Cell Signaling Technology, #7074, Danvers, MA, USA). Chemical inhibitors were used: chloroquine (Sigma-Aldrich, C6628, St. Louis, MO, USA) and MG132 (Sigma-Aldrich, SML1749, St. Louis, MO, USA).

### 4.2. Human Subjects

This study was approved by the Institutional Review Boards at Women & Infants Hospital, Providence, RI. Patients with PE were diagnosed based on ACOG guidelines of preeclampsia (Appendix A). Systolic blood pressures ≥ 140 mmHg or ≥ 160 mmHg and diastolic blood pressure ≥ 90 mmHg or ≥ 110 mmHg measured at or after 20 weeks of gestation were associated with PE or severe PE, respectively. Placental samples were obtained following informed written consent from pregnant women with early-onset PE (e-PE) (< 34 gestational weeks) and gestational age-matched pregnancy. Exclusion criteria included chronic hypertension, gestational or pre-existing diabetes, fetal demise, daily tobacco use, fetal anomalies, and multiple gestations. For placental sample collection, a 1 cm^3^ specimen was removed from the placenta and vigorously washed with chilled phosphate-buffered saline solution. After removing any additional blood from placental tissue, a portion was stored at −80°C until further use. All methods were carried out in accordance with relevant guidelines and regulations.

### 4.3. Cell Culture 

Human primary trophoblasts were isolated from the placental villi at 19 weeks of gestation (purchased from ScienCell Research Laboratories, Carlsbad, CA, USA) and cultured in Trophoblast Medium (TM) (#7121, ScienCell Research Laboratories, Carlsbad, CA, USA) supplemented with 10% FBS, growth factors (ScienCell Research Laboratories, Carlsbad, CA, USA), 100 U/mL penicillin, and 100 µg/mL streptomycin (GIBCO, 15140, Waltham, MA, USA) at 37 °C in a 5% CO_2_ atmosphere. The cells were grown in 10 cm dish, and medium was changed every two days. After full confluency, the cells were split for the experiments.

### 4.4. Hypoxia–Reoxygenation Treatment

Primary human trophoblast (PHT) cells were seeded at 2,000,000 cells in 10 cm dishes and grown in TM medium (ScienCell Research Laboratories, Carlsbad, CA, USA) supplemented with 10% FBS, growth factors (ScienCell Research Laboratories, Carlsbad, CA, USA), 100 U/mL penicillin, and 100 µg/mL streptomycin (GIBCO, 15140, Waltham, MA, USA). After 70% of confluence, cells were washed three times in serum- and growth-factor-free media and grown for 3 days in 5 mL of serum-free media containing only antibiotics under normoxia (21% O_2_) or hypoxia condition. Hypoxia condition was constructed with low oxygen tension (<1% O_2_) using a hermetically enclosed incubator (ProOx Model C21, BioSpherix, Parish, NY, USA) with continuous digital recording of atmospheric oxygen and CO_2_ using a respective sensor connected to a data acquisition module (BioSpherix, Parish, NY, USA). The media were pre-equilibrated with low oxygen tension (<1% O_2_) before addition to the culture plate. On day 3, hypoxia-treated cells were incubated for 3 h under normoxic condition for reoxygenation treatment. The cells were then harvested and lysed in RIPA buffer containing 25 mM Tris-HCl, 150 mM NaCl, 1% sodium deoxycholate, 1% NP40, 0.1% SDS, protease inhibitor cocktail (Roche, 0469316001), and 1% phosphatase inhibitor (Sigma-Aldrich, P5726, St. Louis, MO, USA).

### 4.5. Immunoblotting

Protein concentration was determined using the BCA assay. Equal amounts of protein extracts were separated with 4–15% SDS-PAGE using mini-PROTEAN^®^ TGX Stain-Free™ Precast Gels according to standard procedures (Bio-Rad, Hercules, CA, USA). After being blocked in 5% nonfat milk dissolved in PBS buffer (pH 7.4) containing 0.1% Tween 20 (PBST) for 1 h at room temperature, the transferred PVDF membrane was incubated overnight in primary antibody solution diluted in 5% nonfat milk or 3% BSA in PBST at 4 °C. After several washes, the membrane was incubated for 1h at room temperature with HRP-conjugated donkey anti-rabbit or mouse IgG (1:1000, Cell signaling), treated with chemiluminescence substrate (SuperSignal, ThermoFisher Scientific, Waltham, MA, USA), and exposed on film (Kodak, Rochester, NY, USA). The density of blots was measured using ImageJ (NIH, Bethesda, MA, USA).

### 4.6. Detection of Aggregated Proteins

PHT cells were plated on glass coverslips and grown in TM medium (ScienCell Research Laboratories, Carlsbad, CA, USA). After several washes, the cells were incubated in serum- and growth-factor-free TM medium and then treated with vehicle/chloroquine (50 μM, Sigma-Aldrich, St. Louis, MO, USA) or normoxia/hypoxia–reoxygenation. For detection of aggregated protein, fixed cells were stained with ProteoStat dye using ProteoStat Aggresome Detection Kit (ENZO, ENZ-51023-KP002, Farmingdale, NY, USA) according to the manufacturer’s instruction. Then, coverslips were mounted in VECTASHIELD anti-fade mounting medium containing DAPI (VECTOR LABORATORIE, H-1200, Burlingame, CA, USA). Fluorescence images were captured using a Nikon Eclipse TE2000 (Nikon, Tokyo, Japan) fluorescent microscope and analyzed using MetaVue Imaging software (Molecular Devices, San Jose, CA, USA).

### 4.7. LysoTracker Staining

The LysoTracker Red-99 (Thermo Fisher Scientific, Waltham, MA, USA) was used to detect the biosynthesis and accumulation of lysosomes according to the manufacturer’s instructions. Briefly, the PHTs grown on the coverslips were treated with normoxia or H/R for 3 days. On day 3, normoxia- and H/R-treated cells were incubated in serum-free media containing LysoTracker (at 50 nM) for 20 min at 37 °C. After rinses in DPBS, the cells were incubated for 20 min with fresh serum-free media and then fixed with 4% paraformaldehyde. After extensive washing in PBS, the cells were mounted with anti-fade mounting medium containing DAPI (VECTOR LABORATORIES, H-1200, Burlingame, CA, USA) and observed using a Nikon Eclipse TE2000 (Nikon, Tokyo, Japan) fluorescent microscope and analyzed using MetaVue Imaging software (Molecular Devices, San Jose, CA, USA).

### 4.8. Transmission Electron Microscopy

Ultrathin sections at 70 nm thickness were cut and double-stained with uranyl acetate and lead citrate. For quantification of autophagic vacuoles (autophagosomes and autolysosomes), photomicrographs showing the perinuclear area from hypoxia-treated (*n* = 30) or control cells (*n* = 30) (21,000 magnification) were randomly taken using a Philips 410 transmission electron microscope (Amsterdam, The Netherlands).

### 4.9. Statistical Analysis

Results were presented as the mean ± SEM. Comparisons between experimental groups were performed using a Student’s *t*-test or one-way ANOVA followed by a Bonferroni post hoc test, Fisher’s exact test, or Wilcoxon rank-sum test. Value of *p* < 0.05 was considered statistically significant, whereas *p* > 0.05 not significant (ns).

## Figures and Tables

**Figure 1 ijms-23-05644-f001:**
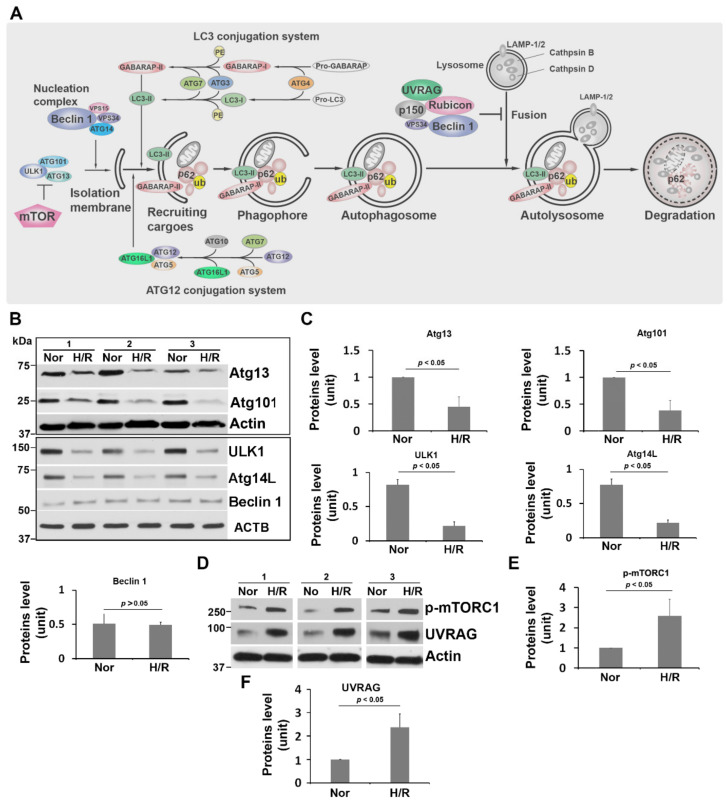
H/R treatment dysregulates the regulators involved in the initiation of isolation membrane formation in PHTs. (**A**) Diagram showing the key molecules involved in regulating the whole processes of autophagic flux. (**B**) Immunoblotting showing the effect of H/R conditions on the abundance of key molecules in ULK1 initiation complex and nucleation complex as shown in (**A**). (**C**) The intensity of target bands shown in (**B**) was measured and statistically compared between normoxia (Nor) and H/R treatment. (**D**–**F**) Western blotting and quantitative analyses of indicated proteins. ACTB (β-actin) was monitored as a loading control among samples. Data are expressed as mean ± SEM and analyzed by a Student’s *t*-test (*n* = 3).

**Figure 2 ijms-23-05644-f002:**
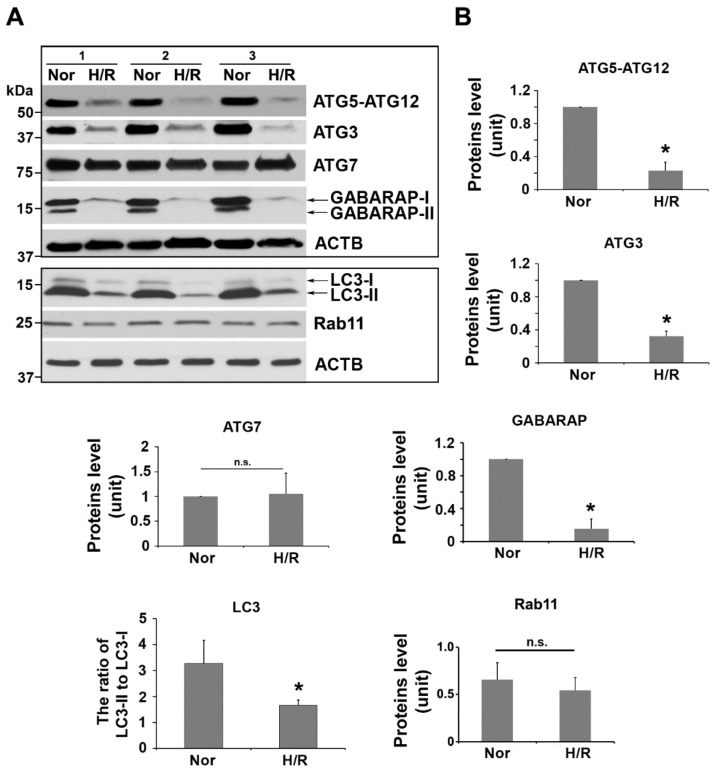
H/R condition disturbs Atg12–Atg5, LC3/GABARAP conjugation systems, and their related regulators in PHTs. (**A**) Western blots of indicated molecules in normoxia (Nor)- and H/R-treated PHTs. (**B**) Quantification of the intensity of bands shown in (**A**). Data are expressed as mean ± SEM and analyzed by a Student’s *t*-test (*n* = 3). *: *p* < 0.05, n.s.: not statistically significant.

**Figure 3 ijms-23-05644-f003:**
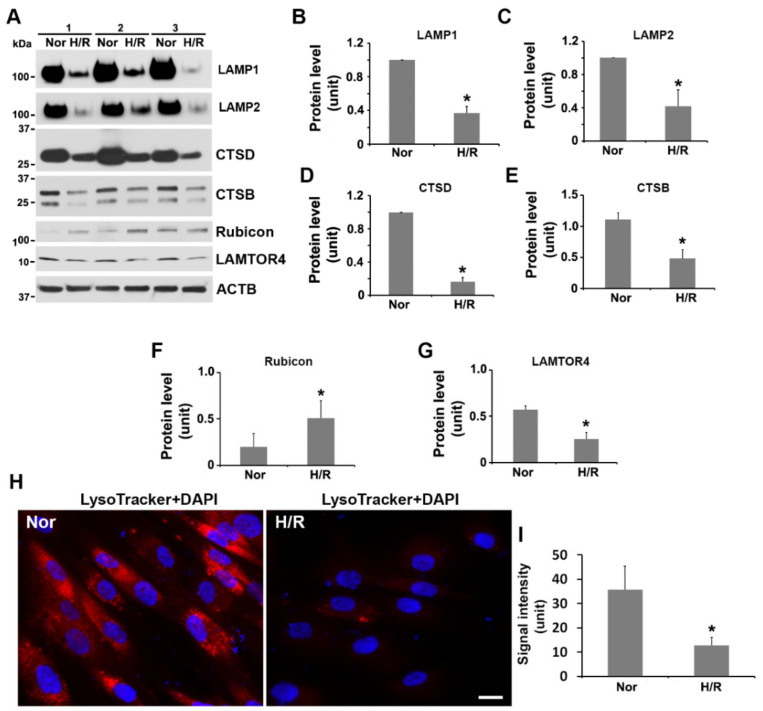
Exposure to H/R impairs lysosomal integrity and activity and autolysosome formation in PHTs. (**A**) The abundance of lysosomal resident proteins and regulators of autolysosome formation in normoxia (Nor)- and H/R-treated PHTs. (**B**–**G**) Quantitative analysis of the band intensity of indicated proteins. (**H**,**I**) LysoTracker staining in normoxia (Nor)- and H/R-treated PHTs (**H**) and quantitative data for comparison of LysoTracker signal intensity between Nor- and H/R-treated cells. The nuclei were stained with DAPI. Data are expressed as mean ± SEM and statistically analyzed by a Student’s *t*-test (*n* = 3, *: *p* < 0.05).

**Figure 4 ijms-23-05644-f004:**
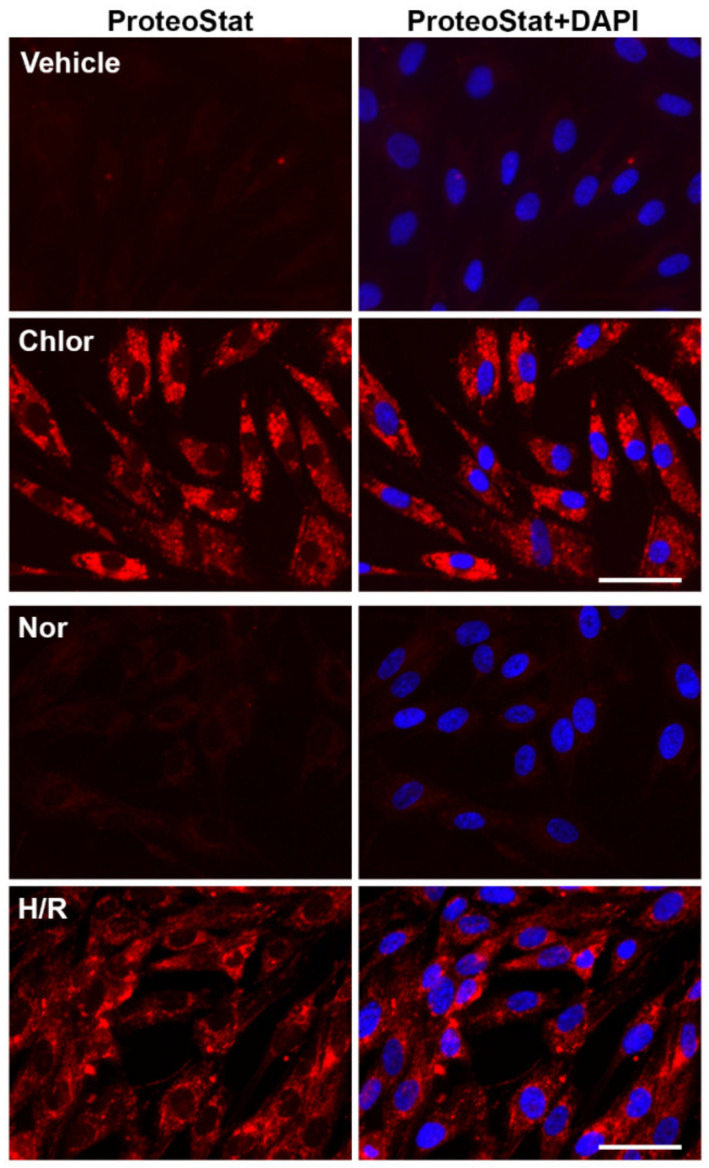
Accumulation of protein aggregates in chloroquine- and H/R-treated PHTs. PHTs were treated with vehicle or chloroquine and fixed at 12 h or exposed to normoxia (Nor) or H/R and fixed at day 3. Fixed cells were stained with ProteoStat dye (red). The nuclei were stained with DAPI (blue). The images were representatives of at least 3 independent experiments. Bar: 20 μm.

**Figure 5 ijms-23-05644-f005:**
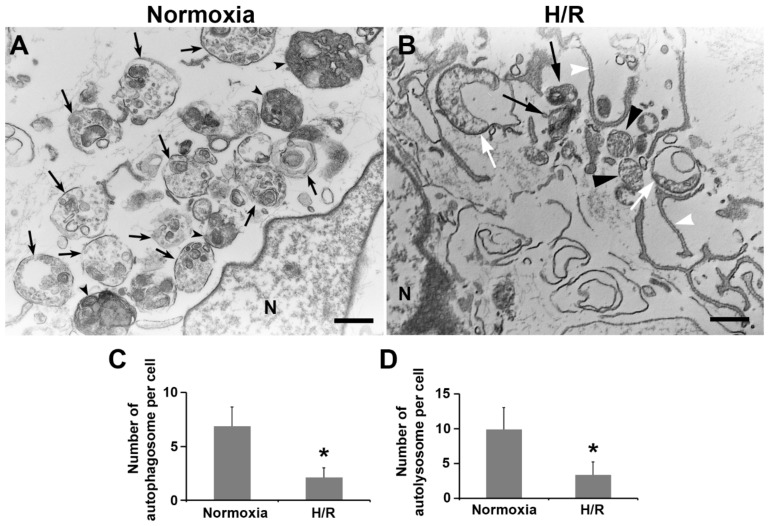
Ultrastructural analysis of autophagic vacuoles in normoxia- and H/R-treated PHTs. (**A**) Representative electron micrograph showing autophagic vesicles in a normoxia-treated cell. Black arrows indicate autophagosomes, while arrowheads show autolysosomes. (**B**) Representative electron micrograph showing autophagic vacuoles in an H/R-exposed cell. White arrows point to damaged mitochondria. White arrowheads indicate isolation membranes. Black arrows and arrowheads show autolysosomes and autophagosomes, respectively. N: the nucleus. Bar: 600 nm. (**C**,**D**) Statistical comparison of the number of autophagosomes and autolysosomes between normoxia- and H/R-treated cells. Data are expressed as mean ± SEM and statistically analyzed by a Student’s *t*-test (*n* = 30, *: *p* < 0.05).

**Figure 6 ijms-23-05644-f006:**
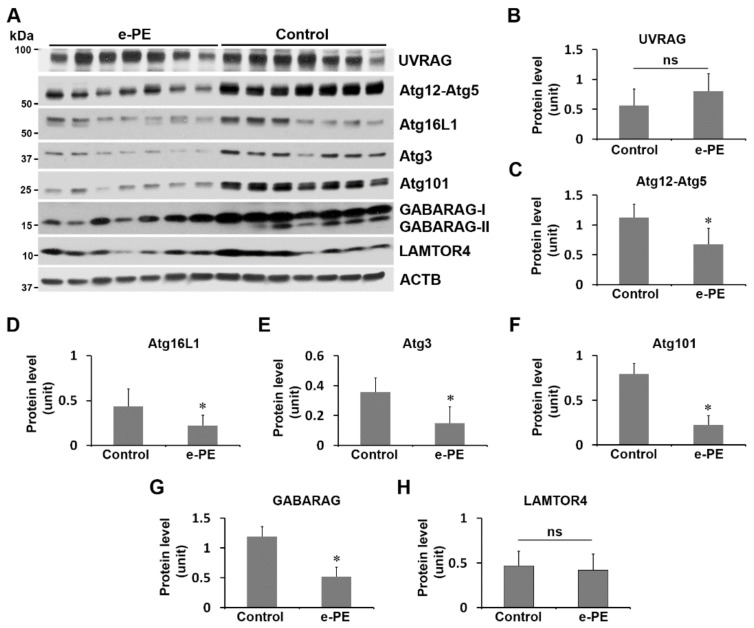
The abundance of key molecules of autophagic flux in the placenta from women with e-PE and gestational age-matched deliveries. (**A**) Placental tissues from e-PE and controls were assessed for protein abundance of indicated proteins using Western blotting. (**B**–**H**) Quantitative data for comparison of the levels of target proteins. Data are expressed as mean ± SEM and statistically analyzed by a Student’s *t*-test (*n* = 7, *: *p* < 0.05).

**Figure 7 ijms-23-05644-f007:**
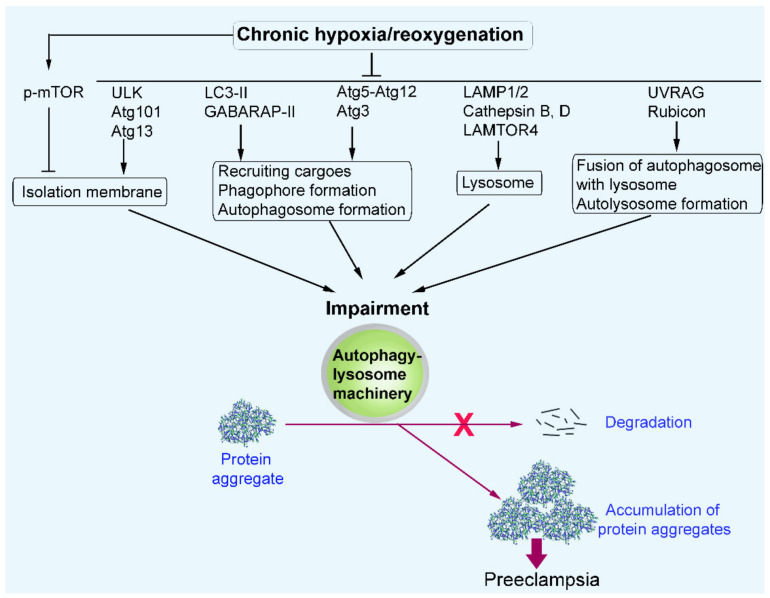
Working model for the effect of chronic hypoxia/reoxygenation on the autophagy–lysosome machinery. Chronic hypoxia/reoxygenation negatively affects the expression of multiple key molecules at each step of the autophagic pathway, leading to impaired autophagy–lysosome machinery in trophoblasts. Impaired autophagy–lysosomal machinery inhibits the degradation of aggregated proteins and subsequently induces the accumulation of protein aggregates in trophoblasts, a mechanism that contributes to the pathophysiology of PE.

## Data Availability

The data presented in this study are available on request from corresponding author.

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
