# Peer review of "Hypoxia-Reoxygenation Impairs Autophagy-Lysosomal Machinery in Primary Human Trophoblasts Mimicking Placental Pathology of Early-Onset Preeclampsia"

_ijms, 2022, doi:10.3390/ijms23105644_

Round 1
Reviewer 1 Report
Cheng S at el. describe the key molecules of the autophagy mechanism that are dysregulated in primary human trophoblasts (PHTs) when submitted to a hypoxia-reoxygenation (H/R) treatment. Autophagy is associated with the pathogenesis of preeclampsia (PE), and using placental tissue of early-onset PE deliveries the authors show a similarly dysregulated expression of the same key regulatory molecules that promote autophagy activity in the placenta.
It is a really interesting work that try to provide new insights about how autophagy is dysregulated in PE, but several minor points should be clarified before publication.
- In the introduction or in the results section it is necessary to describe why the use of hypoxia-reoxygenation (H/R) on primary human trophoblasts (PHTs) is a good cellular model mimicking PE pathophysiology.
- Figure 1 C: Does the quantification of the different protein levels refer to the corresponding loading control levels? In the text the authors comment that ULK1, and Atg14L are significantly down-regulated after H/R stimulation, as well as, the contents of phosphorylated mTORC1 and UVRAG relative to normoxia-treated PHTs significantly augmented. Please, add the p value. Why the quantitative analyses UVRAG is not present in Figure 1C?
- Figure 2: What is the role of Atg 3?
- Page 4, lane 109: “Our results showed a significant decrease in Atg12-Atg5 and Atg6 expression” Where is the WB of Atg6?
- Page 4, lanes 123-125: “Rab11 and found that H/R treatment did not significantly alter the content of this protein when compared to normoxic condition (Figure 2A and I)”. However, and asterisk is on top of the quantification of the intensity of bands (Figure 2B)
- Figure 3: Why the authors show the WB of pro-cathepsin D and cathepsin D? Please explain.
- Page 7, lanes 170-172: “Chloroquine is known to impair lysosomal acidification and consequently disrupt autolysosome formation [18]”. Please add a suitable reference for this statement.
Please, explain why you use chloroquine as a positive control of protein aggregation in primary human trophoblasts.
- Figure 5B: Please add a picture showing some autophagosomes and autolysosomes in H/R-exposed cells.
- Figure 6: Please explain why to conclude whether the autophagy molecular events observed in H/R-stimulated trophoblasts can be replicated in the PE placenta you studied Atg16L1 that was not studied in the trophoblast, but did not studied the other molecules. Why you did not measure mTOR, Beclin, Rab11, Rubicon, LAMP 1 and 2, and cathepsin B and D?
Please, explain the differences found in UVRAG and LAMTOR4 proteins between trophoblast and PE placenta.
Reviewer 2 Report
Primary human trophoblast cells were used to study autophagy-lysosomal machinery to understand mechanism of preclampsia in women. This study is of high priority because use of human material, in vitro system and biochemical methods together with microscopic observation including transmission electron microscopy. Only minor corrections are suggested.
1.The aim of the study should be rewritten. There is mixed information about what was revealed and what was done. There should be information what will be studied based on the background presented in the Introduction
2. Cell culture of trophoblasts needs more details
3. For immunohistochemistry supplier and concentration of used abs should be provided
